# S100A9 Alters the Pathway of Alpha-Synuclein Amyloid Aggregation

**DOI:** 10.3390/ijms22157972

**Published:** 2021-07-26

**Authors:** Zigmantas Toleikis, Mantas Ziaunys, Lina Baranauskiene, Vytautas Petrauskas, Kristaps Jaudzems, Vytautas Smirnovas

**Affiliations:** 1Department of Biothermodynamics and Drug Design, Institute of Biotechnology, Life Sciences Center, Vilnius University, Saulėtekio 7, 10257 Vilnius, Lithuania; mantas.ziaunys@gmc.vu.lt (M.Z.); lina.baranauskiene@bti.vu.lt (L.B.); vytautas.petrauskas@bti.vu.lt (V.P.); vytautas.smirnovas@bti.vu.lt (V.S.); 2Latvian Institute of Organic Synthesis, Aizkraukles 21, LV-1006 Riga, Latvia; kristaps.jaudzems@osi.lv

**Keywords:** S100A9, synuclein, amyloid proteins, fibrils, FTIR, AFM

## Abstract

The formation of amyloid fibril plaques in the brain creates inflammation and neuron death. This process is observed in neurodegenerative disorders, such as Alzheimer’s and Parkinson’s diseases. Alpha-synuclein is the main protein found in neuronal inclusions of patients who have suffered from Parkinson’s disease. S100A9 is a calcium-binding, pro-inflammation protein, which is also found in such amyloid plaques. To understand the influence of S100A9 on the aggregation of α-synuclein, we analyzed their co-aggregation kinetics and the resulting amyloid fibril structure by Fourier-transform infrared spectroscopy and atomic force microscopy. We found that the presence of S100A9 alters the aggregation kinetics of α-synuclein and stabilizes the formation of a particular amyloid fibril structure. We also show that the solution’s ionic strength influences the interplay between S100A9 and α-synuclein, stabilizing a different structure of α-synuclein fibrils.

## 1. Introduction

The formation of protein amyloid fibrils is related to neurodegenerative disorders, such as Alzheimer’s or Parkinson’s disease [1], as well as other amyloidoses [2]. Currently, there are over 30 different amyloid-disease-related proteins and peptides [3], which have both highly distinct amino acid sequences and structural motifs. Wide variety of structures, coupled with the complex mechanism of protein transition from the native state to their aggregated form, has been a major factor limiting the search for potential anti-amyloid compounds [4]. Despite many decades of research, very few treatment modalities are available for amyloid-related disorders [5,6]. Taking into account the ever-increasing number of afflicted patients [7], it is of utmost importance to better understand amyloid protein aggregation.

The intrinsically disordered protein alpha-synuclein (α-syn) is associated with Parkinson’s disease. The N-terminal part of α-syn can fold to an alpha-helical structure after the interaction with lipid surface such as plasma membrane or vesicles. The main physiological function of α-syn in native state, is in trafficking vesicles filled with neurotransmitters in pre-synaptic neuron terminals [8]. However, upon aggregation, α-syn loses the physiological function and is found in large neuronal inclusions (known as Lewy bodies in Parkinson’s disease) or glial cytoplasmic inclusions (in multiple system atrophy) [9,10,11]. Due to the vast number of people affected by Parkinson’s disease, this protein has been the subject of numerous studies in vivo, in vitro and in silico that revealed several peculiar aspects of α-syn aggregation and fibril structure. The first one was the ability of α-syn to form structurally diverse fibrils, which was observed under different conditions in vitro [12,13] or found in brain tissues with different neurodegenerative pathology [14,15]. The second was a highly-stochastic nature of spontaneous aggregation, which caused a relatively high dispersion of the aggregation kinetic curves [16]. Finally, it was also reported that the rate of fibril formation depends not only on the reaction solution pH value [17], ionic strength [18], and protein concentration [19], but it can also be modulated by the presence of other amyloid proteins, such as Tau protein [20], amyloid beta peptide [21], prion protein [22,23,24], S100A9 [25] or non-amyloid proteins like serum albumin [26] and protein chaperon Hsc70 [27].

S100A9 is a calcium-binding protein, mainly expressed by neutrophiles [28] in the response to inflammation or injury [29,30]. Since accumulation of amyloid fibrils is one of the causes for neuroinflammation and S100A9 can form amyloid fibrils in vitro [31] and in vivo [32,33], this may indicate a major role that S100A9 plays in the onset or progression of amyloid-related disorders [34,35]. It has been shown in vitro that when α-syn and S100A9 are both present in the reaction mixture, the fibril formation lag time is considerably reduced, if compared to aggregation of either protein separately [25]. It was also shown, that amyloid plaques contain S100A9 co-localized with amyloid-beta in the brain tissue samples of patients with Alzheimer’s disease [32]. S100A9 can also influence the aggregation properties of amyloid beta in vitro [36]. Despite the reports of such co-interactions, it is still unclear how S100A9 can affect the final structure of other amyloid aggregates. In this work, we examined the aggregation kinetics and final fibril structures of α-syn, when it is aggregated in the presence of different S100A9 concentrations using Fourier transformed infrared (FTIR) spectroscopy and atomic force microscopy (AFM). We also analyzed the aggregation of α-syn under different ionic strength conditions. The study revealed that the aggregation and final structure of α-syn fibrils were affected both by relatively low concentrations of S100A9 and the solution’s ionic strength.

## 2. Results

The aggregation curves of α-syn can be grouped into three distinct populations (Figure 1A), which differ in post-transitional maximum ThT fluorescence intensity. This value increases up to 4, 40 and 80 arbitrary units (a.u.) for the first (A1), the second (A2) and the third (A3) populations, respectively (Figure 1A). The half time of aggregation kinetic curve ($t_{50}$) is also different for these populations: (15 ± 5) h for A1, (23 ± 3) h for A2 and (28 ± 2) h for A3 (Figure 1H).

The maximum fluorescence intensity of aggregation kinetic curves becomes similar to A1, when different concentrations of S100A9 are present in the α-syn sample. Even 2.3 μM S100A9 (which corresponds to 0.033 parts of α-syn concentration) can change the fluorescence intensity to the level of A1 (Figure 1B). The aggregation process becomes the least stochastic upon the addition of 35 μM and 70 μM S100A9 into the α-syn solution. The $t_{50}$ of α-syn with 2.3 μM S100A9 is (15 ± 4) h, the same as for A1. The increased concentration up to 4.5 μM, 9 μM and 18 μM S100A9, shifts $t_{50}$ to (22 ± 3) h, (24 ± 3) h, (21 ± 4) h respectively, which is similar to the control (A2). The presence of 35 μM and 70 μM S100A9 decreases $t_{50}$ to (11 ± 1) h (Figure 1H).

The FTIR spectra of A1, A2 and A3 population fibrils show different secondary structure (Figure 2A,B) in the region which is associated with hydrogen bonding in the β-sheet structure [37]. Population A1 has the main maximum position of FTIR spectrum at 1628 cm^−1^ (the second derivative main minimum at 1626 cm^−1^), while both A2 and A3 have maxima at 1624 cm^−1^ (the second derivative minima at 1623 cm^−1^ and 1624 cm^−1^ respectively), indicating that the A1 population has the weakest hydrogen bonding in the β-sheet structure. The A1 spectrum also has a shoulder at 1636 cm^−1^ reflected by a minimum in the second derivative at 1636 cm^−1^ (weaker type of hydrogen bonding), which is not present in the other two populations. The A2 and A3 population fibrils also have some differences between their secondary structures. This is most evident by the existence of a minimum in the A2 second derivative spectrum at 1641 cm^−1^ and a shoulder at 1614 cm^−1^, which are not present in the case of A3 and indicate the presence of different types of hydrogen bonding. There is also some distinction in the turn/loop motif region, where A2 has a minimum at 1673 cm^−1^, while for A3 it is at 1667 cm^−1^.

In the presence of S100A9 protein, there is no polymorphism in α-syn aggregation. FTIR spectra of α-syn aggregates in the presence of at least 4.5 μM S100A9 look remarkably similar to the spectrum of A1 α-syn conformation with two major minima in the second derivative at 1626 cm^−1^, 1636 cm^−1^ and the shoulder at 1619 cm^−1^, as well as turns-related minimum at 1667 cm^−1^ (Figure 2A,B). Only at the lowest S100A9 concentration (2.3 μM) the spectrum is a bit different with only one broad minimum at 1625 cm^−1^, suggesting a mixture of conformations in the sample. It is important to mention, that only washed pellet, which contained only a trace amount of S100A9 as confirmed by SDS-PAGE (Appendix A Figure A2), was used to collect FTIR data. To determine that the observed effect is related to S100A9 and not to the presence of any similar size protein, the aggregation experiment was carried out with 70 μM
α-syn and 70 μM hen egg-white lysozyme (14.4 kDa). Unlike with 70 μM S100A9, three populations with distinct ThT fluorescence intensity and FTIR spectra (Appendix A Figure A3) remain similar to the control sample without S100A9.

The AFM images show that the amyloid fibrils are formed in all cases independent on S100A9 concentration (Figure 3).

More clustered fibrils are observed by increasing concentration of S100A9. Cross-sectional height analysis of fibrils in AFM images shows that the mean values of A1 and A2 fibril heights are (5 ± 1) nm, but this value is (7 ± 1) nm for A3 (Figure 3A–C). The fibril-cross-sectional mean heights increase from (6 ± 1) nm to (8 ± 2) nm by increasing S100A9 concentration from 2.3 μM to 18 μM and decrease down to (7 ± 2) nm for the sample with 35 μM and 70 μM S100A9 (Figure 3J).

In order to examine the effect that ionic strength has on this interaction, the aggregation reaction was conducted under multiple NaCl concentration conditions. For the experiments, 35 μM of S100A9 was used, as it resulted in both low $t_{50}$ value deviations (Figure 1F,H), as well as a stable α-syn fibril conformation (Figure 2). The addition of 50 mM NaCl to the α-syn solution increases fluorescence intensity and $t_{50}$ of aggregation curves (Figure 4B,G) compared to the control (Figure 4A,G) and stabilizes the population with higher fluorescence intensity (similar to A3). The curves become less stochastic and shift to the left (shorter $t_{50}$), as the concentration of NaCl is increased from 125 mM to 375 mM. The addition of NaCl to α-syn sample with S100A9 does not change the aggregation kinetic curves and $t_{50}$ significantly, (Figure 4D–G), except the sample with 375 mM NaCl, which has 2–8 times higher post-transitional fluorescence intensity (Appendix A Figure A1).

When additional 50 mM NaCl is present in the aggregation reaction mixture, the profile of FTIR spectrum and the second derivative are very similar to the one of A3 population (Figure 5A,B). A further increase in NaCl concentration has minimal effect on the FTIR spectra, apart the minima at 1640 cm^−1^ and the more expressed shoulder at 1614 cm^−1^, indicating a minor change of hydrogen bond strength.

When the samples for aggregation are composed of 70 μM
α-syn, 35 μM S100A9 and different concentrations of additional NaCl, the presence of 50 mM or 125 mM NaCl does not affect the secondary structure of the resulting fibrils (Figure 5C,D). The sample aggregated in the presence of additional 250 mM NaCl has the main maximum of the spectrum and the main minimum of the second derivative spectrum at 1623 cm^−1^ and a shoulder appears at 1616 cm^−1^, indicating the formation of stronger hydrogen bonds. There is also a minimum at 1670 cm^−1^ associated with the fibril turn/loop motifs. At the highest NaCl concentration, the main minimum remains at a similar position and we observe a loss of the 1616 cm^−1^ shoulder. There is also a small variation in the turn/loop motif region, which becomes more similar to the lower NaCl concentration samples. The FTIR spectrum of α-syn aggregated in the presence of S100A9 and 375 mM NaCl is very similar to the spectrum of A3 population.

AFM analysis shows that the fibrils become more clustered at increasing NaCl concentrations, when aggregation was conducted with or without 35 μM of S100A9 (Figure 6). The cross-sectional height of fibrils in α-syn and α-syn with S100A9 samples slightly decrease at 50 mM and start to increase at 125 mM to 375 mM concentrations of NaCl (Figure 6I). The cross-sectional heights become more variable at increasing concentrations of NaCl.

## 3. Discussion

The results of α-syn aggregation in the absence of S100A9, showed the existence of three different populations of amyloid fibrils. The most notable distinction is different FTIR spectra, showing variation in β-sheet structure hydrogen bonding and some differences in loops and turns. The ThT-binding characteristics for these populations are also different, where we observe 10-fold difference in the maximum fluorescence intensity values at the end of aggregation for A2 and A3 compared to A1. The $t_{50}$ values are also population-specific, with A1 fibrils requiring the shortest amount of time to form and A3 aggregation takes the longest time. AFM images of all three populations do not portray any significant variation in fibril length, however, the A3 aggregates have a significantly larger height, if compared to both other populations. While this type of aggregation forming different populations of fibrils was reported for prion proteins [38], it was not observed for alpha-synuclein [25] under similar conditions as were used in this work. This may be due to the lower number of repeats in the aforementioned study, which did not allow to observe a clear population distribution. It may also be due to a lower concentration of ThT (20 μM) as opposed to 50 μM in these experiments, which may impede the detection of higher ThT fluorescence aggregates.

This massive variation in fibril types, however, is offset by the addition of even a small concentration of S100A9, where both the $t_{50}$ values, as well as fluorescence intensity become similar to the A1 population. These results together with the FTIR spectra show that S100A9 can stabilize the formation of a certain type of α-syn aggregates. Interestingly, the AFM images show that α-syn aggregated in a presence of S100A9, forms thicker fibrils with an average height similar to the A3 population. This observation is in contrast to both the FTIR and ThT fluorescence results. One possibility is that the interaction between α-syn and S100A9 causes a higher level of lateral association of the fibril fibers, thus increasing their average height, while retaining the same secondary structure and ThT binding properties as the A1 population. The interaction of α-syn with S100A9 can also influence the rate of fibril formation, suggesting that α-syn and S100A9 interact prior to nuclei formation or during it. The effect on $t_{50}$ is most substantial at lower S100A9 concentrations (from 2.3 μM to 18 μM), after which it settles at a specific value, indicating a saturating effect. While in the case of FTIR spectra, the structural transition occurs between 0 μM and 4.5 μM S100A9 and then remains virtually unchanged at higher S100A9 concentrations.

In order to examine whether the interaction between α-syn and S100A9 depends on ionic strength, the aggregation experiments were repeated under different concentrations of NaCl. Our results showed that higher concentrations of NaCl and 35 μM S100A9 accelerate the aggregation kinetics. However, matters become much more complicated when we examine the differences in structural aspects. In the absence of S100A9, the FTIR spectra show that the presence of additional NaCl stabilize the structure of A3 population. If α-syn samples contain S100A9, the secondary structure of α-syn fibrils remains stable up to 125 mM. The further increase of NaCl concentration causes a shift, which makes the FTIR spectrum similar to that of S100A9-free sample populations A2 or A3. These results point towards the conclusion that α-syn and S100A9 interactions are only significant at lower ionic strength conditions and that the interplay between both proteins is, in fact, at least partially dependent on ion-pair formation.

The interaction dependence on ionic strength could be possible due to the different total negative charge of α-syn (pI 4.7) compared to S100A9 (pI 5.7). Moreover, α-syn has an exceptionally negatively charged C-terminal region, which might be a site that interacts with positively charged amino acid residues of S100A9, which cluster on one face of the protein structure. It is reported that the negatively-charged C-terminus has an important influence on the aggregation of α-syn [39,40,41]. Furthermore, truncated sequence of α-syn was found in the neuronal inclusions of patients who have suffered from dementia with Lewy bodies [42]. Human serum albumin was also shown to change the aggregation kinetics of α-syn, the most likely due to serum albumin interaction with the C-terminal region of α-syn [26].

Considering the shared localization in vivo, as well as reports of both S100A9 and α-syn proteins found aggregated in amyloid plaques, this protein interplay may have a major role in the onset and progression of amyloid-related neurodegenerative disorders. This can manifest in multiple distinct or interconnected ways. If S100A9 alters the rate of α-syn aggregation in vivo, then it may induce an earlier onset of neurodegeneration. The resulting different population of α-syn aggregates may also have distinct propagation or toxicity levels, which could increase the rate of neuronal cell death.

## 4. Materials and Methods

### 4.1. Protein Production

The expression of α-syn was performed in *E. coli* BL21(DE) one star strain transformed with the plasmid pRK172 containing human α-syn cDNA as it was used previously [43]. The culture was growing in ZYM-5052 auto-inducing growth media [44] in shaking flasks, 220 RPM, 30 ${}^{{}^{\circ}}C$ for 12 h. The harvested cells were re-suspended in 20 mM Tris-HCl buffer, 1.0 mM EDTA, 2.0 mM 1,4-dithio-d-threitol (DTT), 2.0 mM phenylmethanesulfonyl fluoride (PMSF) protease inhibitor, 500 mM NaCl, pH 8.0. The suspension was sonicated 10 min (pulses of 2 s sonication, 2 s break) and the lysate centrifuged for 30 min, 20,000× *g* at 4 ${}^{{}^{\circ}}C$. The supernatant of lysate containing α-syn was incubated in 90 ${}^{{}^{\circ}}C$ heated water bath until the temperature in the lysate supernatant reached 80 ${}^{{}^{\circ}}C$ and kept under those conditions for 10 min. The soluble poteins were separated from the aggregated ones by centrifugation for 30 min, 20,000× *g* at 4 ${}^{{}^{\circ}}C$. The supernatant was saturated with 40% ammonium sulfate on ice, pellets with α-syn were collected by centrifugation, dissolved in 20 mM Tris-HCl buffer, pH 8.0, dialysed against the same buffer for 1 h and loaded on Q Sepharose FF (GE Healthcare, Bio-Sciences AB, Uppsala, Sweden) 20 mL column. Pure protein was eluted at approximately 300 mM NaCl concentration of the applied linear gradient. The protein was dialysed against water, concentrated up to 500 μM, frozen at −80 ${}^{{}^{\circ}}C$.

The construct to express S100A9 was used as described previously [45]. The expression and preparation of cell-lysate of S100A9 was done the same as described for α-syn, except the lysis buffer, which was without NaCl. The supernatant of cell-lysate was saturated with 70% ammonium sulfate solution on ice, pellets were separated by centrifugation, the supernatant was dialysed against 20 mM Tris-HCl buffer, 500 μM EDTA, 500 μM DTT, pH 8.0 for 3 h and loaded on Q Sepharose FF (GE Healthcare, Bio-Sciences AB, Uppsala, Sweden) 20 mL column. S100A9 was eluted at approximately 250 mM NaCl concentration of the applied linear gradient. The most pure fractions containing S100A9 were concentrated using spin filters (Amycon, 10 kDa MWCO) and purified by gel filtration using 20 mL Superdex 75 (GE Healthcare, Bio-Sciences AB, Uppsala, Sweden) column and 200 mM of ammonium bicarbonate as elution buffer. The most pure S100A9 fractions were freeze-dried and kept at −80 ${}^{{}^{\circ}}C$.

### 4.2. ThT Assay

Protein aggregation kinetics by ThT assay [46] were performed in 96-well Corning non-binding half-area plate with one glass bead (diameter 3 mm) in each well. The plate, filled by 80 μL of protein solution in each well, was shaking constantly between readings (every 10 min) using 300 RPM orbital agitation in CLARIOstar Plus (BMG Labtech, Ortenberg, Germany) plate reader at 37 ${}^{{}^{\circ}}C$ for 60 h. Sample fluorescence intensity was measured using excitation and emission wavelengths of 440 nm and 480 nm respectively. All protein samples contained 70 μM
α-syn, 50 μM ThT, 0.02% NaN_3_, PBS buffer (10 mM Na_2_HPO_4_, 1.8 mM KH_2_PO_4_, 137 mM NaCl, 2.7 mM KCl), pH 7.4. Other samples contained 2.3 μM, 4.5 μM, 9.0 μM, 18 μM, 35 μM, 70 μM of S100A9 (during experiments on how S100A9 influences aggregation of α-syn). For a negative control sample, 70 μM of hen egg-white lysozyme prepared from powders (Sigma-Aldrich, St. Louis, MO, USA, cat. No. L6876) was used in a place of S100A9. Samples with different NaCl concentrations contained 35 μM S100A9 and additional 50 mM, 125 mM, 250 mM, 375 mM NaCl. The $t_{50}$ was determined by fitting the sigmoidal curve to aggregation kinetic data.

### 4.3. FTIR Spectroscopy

Multiple repeats of each sample were combined and centrifuged at 12,500 RPM for 20 min, the supernatant was removed and the fibril pellet was resuspended into 500 μL D_2_O (with 400 mM NaCl to improve sedimentation [47]. After repeating the centrifugation and resuspension procedure three times, the fibril pellet was resuspended into a final volume of 150 μL. For each sample, 256 interferograms with 2 cm^−1^ resolution were collected using Bruker Invenio S spectrometer equipped with a liquid-nitrogen-cooled mercury-cadmium-telluride detector at room temperature in a dry-air flow-through chamber. A D_2_O spectrum was subtracted from each sample spectrum, after which the spectra were baseline corrected in the region at 1595 cm^−1^ to 1700 cm^−1^ and normalized. All data processing was done using GRAMS software.

### 4.4. Atomic Force Microscopy

The AFM images were recorded at high resolution (1024×1024 pixels per 10 μm image as described previously [48]. The samples after aggregation kinetic experiment were diluted 20 times with MilliQ water and 30 μL of this was applied on mica surface freshly modified with 3-(Triethoxysilyl)propan-1-amine (APTES) (Sigma-Aldrich, Co., MO 63103, USA) [49]: 30 μL of 0.5% APTES solution in water is applied on the freshly activated mica surface, incubated for 10 min, washed with 2 mL of water and dried under gentle air-flow. The sample was incubated 10 min on this APTES-mica surface, washed with 1 mL of water and dried applying gentle air-flow. The images were analyzed using Gwyddion software [50]. The heights of the fibrils were determined by fitting a Gauss function to the cross-sectional peaks of each fibril.

### 4.5. SDS-PAGE

The pellet (fibrils) fraction for the gel was prepared by taking 3 μL of the fibril sample prepared for FTIR analysis, diluting it 5 times with 12 μL 10 M urea and mixing with 5 μL 4 times concentrated SDS-PAGE sample buffer containing 40 mM DTT. The samples for testing soluble fraction after aggregation were prepared by taking 15 μL of supernatant after the first centrifugation described in method section “FTIR spectroscopy” and mix it with 5 μL 4 times concentrated SDS-PAGE sample buffer containing 40 mM DTT. A portion of soluble samples containing α-syn with 35 μM and 70 μM S100A9 were filtered through 0.22 μm regenerated cellulose filters and the filtrates prepared for the analysis the same way as soluble fraction samples. The α-syn and S100A9 control samples before aggregation were prepared the same way as soluble fraction samples. All samples were heated at 95 ${}^{{}^{\circ}}C$ for 10 min and 8 μL of each sample was loaded on 12% acrylamide gel.

## 5. Conclusions

The results of this work show that even small concentrations of S100A9 have a major influence on both the formation of α-syn fibrils, as well as their secondary structure and morphology. In addition, the ionic strength is important for S100A9 interaction with α-syn. S100A9 and high ionic strength can change the pathway of α-syn fibril formation, favoring a particular structure from the ensemble of structural populations.

## Figures and Tables

**Figure 1 ijms-22-07972-f001:**
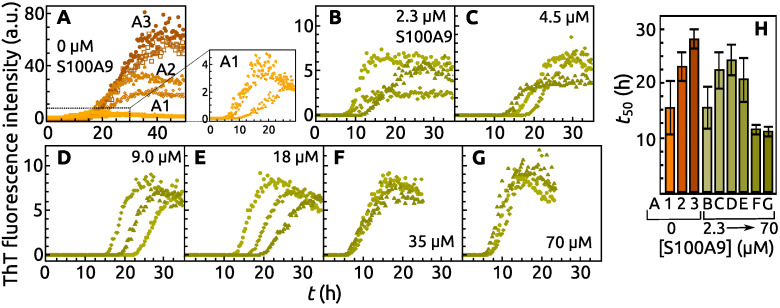
Aggregation kinetic curves of α-syn (**A**–**G**) and the respective $t_{50}$ (**H**) at different concentrations of S100A9: 0.0 μM (**A**), 2.3 μM (**B**), 4.5 μM (**C**), 9.0 μM (**D**), 18 μM (**E**), 35 μM (**F**), and 70 μM (**G**). A1, A2 and A3 denote different populations of α-syn kinetic curves. Protein solution: 70 μM
α-syn, PBS buffer, 0.02% NaN_3_, 50 μM ThT, pH 7.4. Aggregation performed at 37 ${}^{{}^{\circ}}C$ with constant agitation of 300 RPM. The error bars are the standard deviation from the mean (12 repeats).

**Figure 2 ijms-22-07972-f002:**
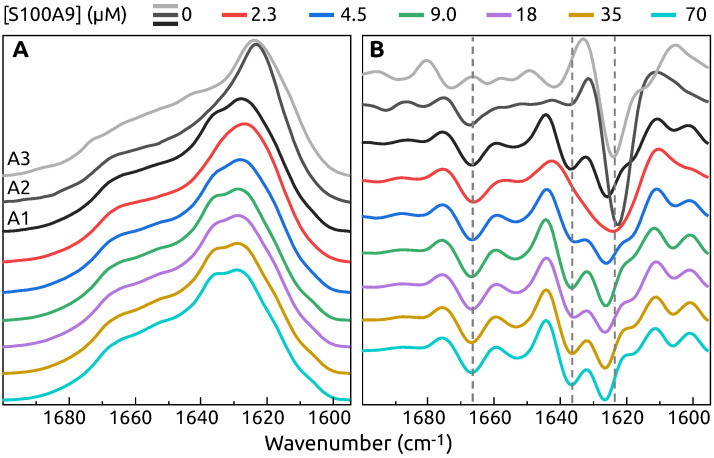
FTIR spectra (**A**) and the second derivatives (**B**) of fibrils obtained after α-syn aggregation with different concentration of S100A9: 0.0 μM, 2.3 μM, 4.5 μM, 9.0 μM, 18 μM, 35 μM, 70 μM. The spectra and second derivatives of α-syn fibril populations are indicated by A1, A2 and A3. Concentration of S100A9 is color coded. Fibrils formed in PBS buffer, 50 μM ThT, 0.02% NaN_3_, 300 RPM at 37 ${}^{{}^{\circ}}C$ for 60 h.

**Figure 3 ijms-22-07972-f003:**
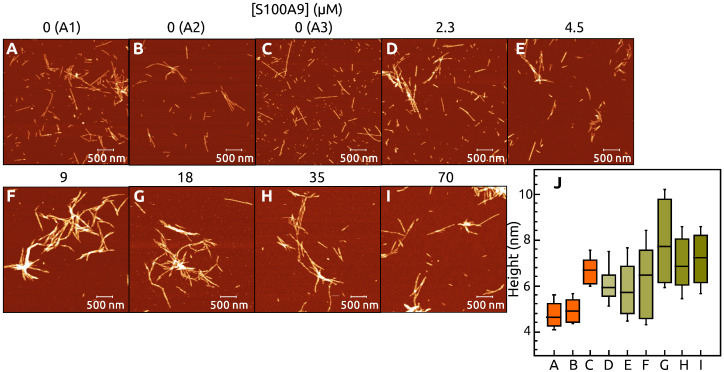
AMF images of α-syn (70 μM) aggregated without (**A**–**C**) or with different concentration of S100A9: 2.3 μM (**D**), 4.5 μM (**E**), 9 μM (**F**), 18 μM (**G**), 35 μM (**H**) and 70 μM (**I**). The cross-section height of fibrils (**J**). The interquartile region (shown by the box), standard deviation (whiskers) and a median (horizontal line inside the box) were calculated from 50 fibrils.

**Figure 4 ijms-22-07972-f004:**
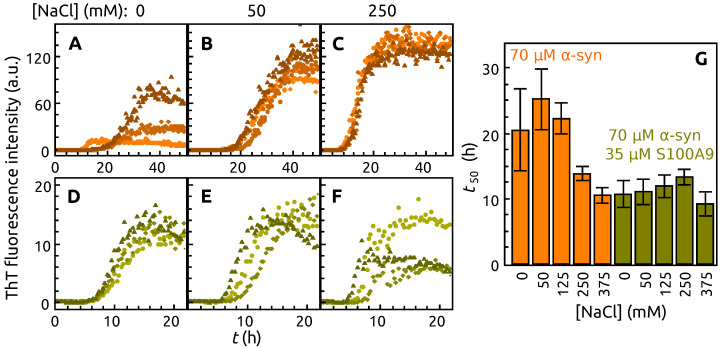
Aggregation kinetics of α-syn without (**A**–**C**) and with S100A9 (**D**–**F**) in a presence of different NaCl concentrations: 0 mM (**A**,**D**), 50 mM (**B**,**E**), 250 mM (**C**,**F**). Each sample represented by 3 mostly recurring curves. The $t_{50}$ value (determined from 12 repeats) of α-syn (orange) and α-syn aggregated with S100A9 (olive) at different concentrations of NaCl (**G**). Protein solution: 70 μM
α-syn, 0 μM or 35 μM S100A9, 50 μM ThT, 0.02% NaN_3_, PBS buffer, pH 7.40.

**Figure 5 ijms-22-07972-f005:**
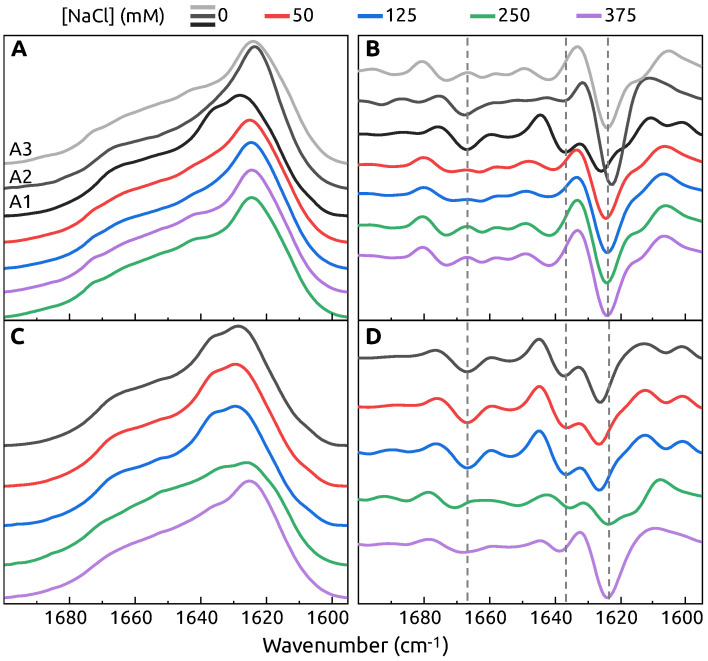
FTIR spectra (**A**,**C**) and second derivative (**B**,**D**) of α-syn (70 μM) without or with 35 μM S100A9 in PBS buffer with different concentration of NaCl.

**Figure 6 ijms-22-07972-f006:**
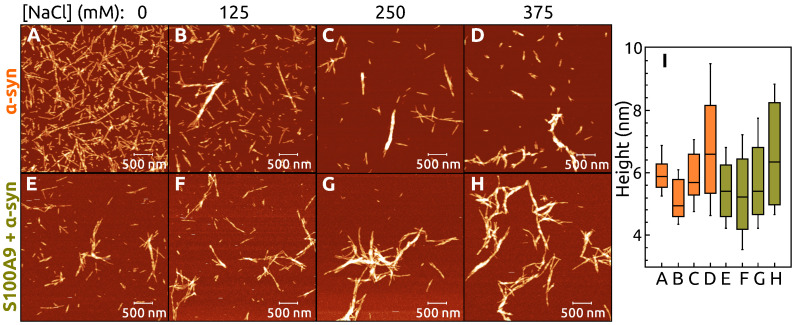
AMF images of α-syn aggregated without (**A**–**D**) and with (**E**–**H**) S100A9 in the presence of different concentrations of NaCl (additional to PBS): 0 mM (**A**,**E**), 125 mM (**B**,**F**), 250 mM (**C**,**G**), and 375 mM (**D**,**H**). Panel (**I**) shows the heights of fibrils in the mentioned conditions from (**A**) to (**H**). The boxes, lines inside them, and whiskers correspond to interquartiles, medians, and standard deviations of the height distribution, respectively. Protein solution: 70 μM
α-syn, 0 or 35 μM S100A9, 50 μM ThT, PBS, additional NaCl as mentioned, pH 7.4.

## Data Availability

All data can be provided upon request.

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
