# Peer review of "S100A9 Alters the Pathway of Alpha-Synuclein Amyloid Aggregation"

_ijms, 2021, doi:10.3390/ijms22157972_

Round 1
Reviewer 1 Report
The authors have analyzed in detail the interaction between α-synuclein and S100A9 and have well documented its biochemical properties. The manuscript is well written, and the results are very convincing.
Unfortunately, there are no data on the toxicity mechanism using cultured cells or mouse models. Thus, the physiological significance of the formed α-synuclein-S100A9 complex in the process of neurodegeneration is unknown, and how it may exert toxicity has not been documented well.
Minor point
The authors should describe the possible physiological significance of the α-synuclein-S100A9 complex in the pathogenesis of α-synucleinopathy including Parkinson's disease.
Author Response
We agree that it would be interesting to examine the toxicity of the α-synuclein-S100A9 complex in future works, however, the manuscript’s main focus is on the effect that S100A9 has on α-synuclein aggregation and the resulting fibril structure in vitro.
We have added a statement at the end of the discussion section to reflect the possibility of the alpha-synuclein-S100A9 complex having distinct propagation or toxicity levels, which may alter the pathogenesis of neurodegenerative disorders.
Reviewer 2 Report
The manuscript entitled "S100A9 Alters the Pathway of Alpha-Synuclein Amyloid Aggregation", by Toleikis and coworkers, is an interesting study on the influence of the protein S100A9 on the fibrillation process of Alpha-Synuclein. The authors employ spectroscopic and microscopic methods to assess how different concentrations of S100A9 at different ionic strengths alter the formation of Alpha-Synuclein amyloid aggregates.
The manuscript is interesting and well written, and a rigorous methodological approach has been employed. This reviewer, however, thinks that additional interventions are necessary before publication, in order to improve the effectiveness of the message and the overall quality of the paper.
Major points
1) The major point regards the lack of negative controls in the experimental design. In order to claim that the described effects on Alpha-Synuclein aggregation are specifically due to the presence of S100A9, I suggest to repeat the experiments with another protein of similar size (at least the set at 70 uM concentration).
2) FTIR spectroscopy can distinguish intra- from inter-beta-sheets, highlighting the formation of amyloid aggregates. The authors should specify and comment the nature of the beta-sheets detected in the FTIR experiments.
3) ThT fluorescence and FTIR spectroscopy indicate that S100A9 promotes an aggregation kinetics and a secondary structure similar to A1, while AFM shows that S100A9 induces morphological features (i.e. fibril thickness) similar to A3. A deeper discussion on the interpretation of these differences should be usefull to the reader.
Minor points
1) The color code in Figure 2 is missing
2) Line 107: 50 uM NaCl is probably 50 mM NaCl
Author Response
1) Regarding your suggestion, we have repeated the aggregation experiment with 70 μM concentration of lysozyme instead of S100A9, as it has a similar molecular weight. The results show that the population dispersion remained and we observed three different ThT fluorescence intensity groups with distinct FTIR spectra (similar to the control sample). We have added this data as an Appendix Figure and referenced it in the result section.
2) With our best knowledge there is no absolute way to distinguish intra- from inter-beta-sheets just by FTIR spectra. Usually, we can safely talk about intra- and inter-molecular beta-sheets only when we have amyloid formation from natively beta-sheet-containing protein and a clear shift in amide I wavenumber is observed upon aggregation. Usually native beta-sheets reflect in higher wavenumbers (1620-1640 cm-1) and intermolecular – in lower wavenumbers (<1620 cm-1), but it is not an absolute rule, especially in amyloid aggregates, where we quite often can observe FTIR amide I band between 1620-1630 cm-1, but there is no evidence of intramolecular beta-sheets while probed with high resolution methods (such as ssNMR). Due to this reason, while talking about FTIR spectra of amyloid aggregates we assume all beta-sheets to be intermolecular, unless there is an opposite evidence from other methods.
3) We have expanded the discussion regarding the point on how the thicker fibrils may be a result of different fiber lateral association tendencies due to the presence of S100A9.
Minor points were solved:
1) We have modified Figure 2 to indicate which color corresponds to which concentration.
2) We have fixed the error in line 107.
Reviewer 3 Report
The study “S100A9 Alters the Pathway of Alpha-Synuclein Amyloid Aggregation” by Toleikis et al. describes how alpha-synuclein aggregation kinetics changes in the presence of pro-inflammatory protein S100A9, as well as in the presence of different concentrations of NaCl. Using the combination of ThT fluorescence spectroscopy, FTIR, and AFM, the authors found that S100A9 accelerates the aggregation of alpha-synuclein and shifts the morphology of aggregates from poly to monomorphic structure. The structure of amyloid fibrils was altered by increasing ionic strength, as well, which brings to a conclusion that there are important electrostatic interactions between alpha-synuclein and S100A9.
In my opinion, the study is carefully designed, the methodology was appropriately chosen and the conclusions are supported by the presented data.
I only have a few minor concerns:
- The authors should explain the discrepancy in their results compared to the previous study done on the same system by Horvath et al. (ref. 25 in the manuscript), i.e. why do they see three different populations of alpha-synuclein fibrils in comparison to the ref. 25?
- Line 107: 50 uM should be 50 mM
- Figure 2: Indicate the concentrations within the Figure (not just by writing that it is color-coded)
- Explain why 35 uM concentration of S100A9 was chosen for the ionic strength studies
- Line 165: shows - show
Author Response
-
The authors should explain the discrepancy in their results compared to the previous study done on the same system by Horvath et al. (ref. 25 in the manuscript), i.e. why do they see three different populations of alpha-synuclein fibrils in comparison to the ref. 25?
We have expanded the discussion to address this point. The alpha-synuclein aggregation experiments in the manuscript by Horvath et al. were done with 5 repeats, as opposed to 12 in our manuscript. It is very possible that there were not enough repeats to observe the random appearance of these populations. The concentration of ThT was also lower, which may have limited the detection of a high-fluorescence intensity population. We have noticed this population distribution prior to the experiments in our manuscript, which is why a larger number of repeats was used.
-
Line 107: 50 uM should be 50 mM
We have fixed the error in line 107.
-
Figure 2: Indicate the concentrations within the Figure (not just by writing that it is color-coded)
We have modified Figure 2 to indicate which color corresponds to which concentration.
-
Explain why 35 uM concentration of S100A9 was chosen for the ionic strength studies
We have added an explanation at the start of the ionic strength study section. The concentration was chosen, because it resulted in low t50 value deviations and a stable α-syn fibril conformation.
-
Line 165: shows – show
We have fixed the error in line 165.
Round 2
Reviewer 2 Report
The authors replied to all the reviewer's concerns, performing the suggested experiments. I therefore recommend publication.